# Epidemiology of HIV in Remote Equatorial Regions of Cameroon: High Prevalence in Older Adults and Regional Disparities

**DOI:** 10.3390/tropicalmed10120334

**Published:** 2025-11-27

**Authors:** Marcel Tongo, Yannick F. Ngoume, Ramla F. Tanko, Urmes C. Teagho, Brice Eselacha, Oumarou H. Goni, Dell-Dylan Kenfack, Mérimé Tchakoute, Georges Nguefack-Tsague

**Affiliations:** 1Centre of Research for Emerging and Re-Emerging Diseases/Institute of Medical Research and Studies of Medicinal Plants (CREMER/IMPM), Yaounde P.O. Box 906, Cameroon; mkfranck.2.08@gmail.com (Y.F.N.); tramlaf@gmail.com (R.F.T.); urmesteagho@gmail.com (U.C.T.); gonioumar@outlook.fr (O.H.G.); dellkenfack@gmail.com (D.-D.K.); 2Higher Institute for Scientific and Medical Research (ISM), Yaounde P.O. Box 5797, Cameroon; 3HIV Pathogenesis Programme, The Doris Duke Medical Research Institute, University of KwaZulu Natal, Private bag 7, Congella, Durban 4013, South Africa; 4National Institute of Cartography (INC), Ministry of Scientific Research and Innovation, Yaounde P.O. Box 157, Cameroon; eselachabrice94@gmail.com; 5Programmes de Santé et Développement au Sein du Groupement de la Filière Bois du Cameroun (GFBC), Douala P.O. Box 1605, Cameroon; merimetchakoute@yahoo.fr; 6Faculty of Medicine and Biomedical Sciences, The University of Yaoundé I, Yaounde P.O. Box 1364, Cameroon; nguefacktsague@gmail.com

**Keywords:** HIV prevalence, equatorial rainforest, Cameroon, socio-demographic factors

## Abstract

Data on HIV prevalence in remote, rural communities of Cameroon are scarce despite the country’s high HIV-1 group M diversity. This study assessed HIV seroprevalence and socio-demographic determinants in four regions of the equatorial rainforest location of the country. A cross-sectional survey was conducted among 5631 individuals in rural communities of the Centre, East, Littoral, and South regions. HIV testing was performed, and socio-demographic data were collected. Prevalence ratios (PRs) and adjusted prevalence ratios (aPRs) were estimated using bivariate and multivariate analyses (negative log-binomial model with generalised estimating equations, GEEs). Overall, HIV prevalence was 3.4% (95% CI: 2.9–3.9%) in individuals aged 15–49 years and 4.9% in those aged ≥50 years. Women had higher prevalence than men (4.5% vs. 3.0%, aPR = 1.53, 95% CI: [1.12–2.08], *p* = 0.007) and also higher HIV prevalence among individuals aged 50–54 years compared to those aged 15–19 years (5.5% vs. 1.8%, aPR = 2.76, 95% CI: [1.24–6.15], *p* = 0.013). The South region recorded the highest prevalence (5.2%, aPR = 1.82, 95% CI: [1.04–3.18], *p* = 0.035) compared to the Centre region with the lowest (2.3%). Divorced/separated/widowed individuals (10.2%) had increased risk (aPR = 1.70, 95% CI: [0.80–3.58], *p* = 0.165) compared to single individuals (3.2%). HIV remains a significant public health concern in remote, rural Cameroon, with a disproportionate impact on older adults and women. Surveillance should extend beyond the traditional 15–49-year age range, and targeted prevention is needed for high-prevalence regions and older populations to curb ongoing transmission.

## 1. Introduction

According to the latest data from the Joint United Nations Programme on AIDS (UNAIDS), there were 1.3 million new HIV infections in 2024 [1]. Most of these infections happened in sub-Saharan Africa, with women and girls accounting for 44% [2]. Cameroon has an estimated national adult HIV prevalence between 2.5% and 3.8% [3], with marked disparities across regions and sexes. However, existing surveillance systems primarily focus on urban areas and the 15–49-year age group, leaving limited data on rural populations and adults aged 50 years and above. In addition, the HIV epidemic in the country is generalised, meaning it is not primarily restricted to high-risk population groups. Furthermore, studies of HIV prevalence in the country have been limited and have typically focused either on specific high-risk population groups or residents of major cities. For example, in the early 1990s, HIV infection rates in Yaoundé were 1.2%, 2.2%, 2.6%, and 7.6% among blood donors, pregnant women, patients with sexually transmitted diseases (STDs), and patients with tuberculosis (TB), respectively [4,5]. Prevalence rapidly increased thereafter, reaching 21.6% in TB patients and 34% among female sex workers (FSW) by 1996 [6,7].

In 2000, we assessed HIV prevalence in certain rural communities across four administrative regions of the equatorial rainforest of Cameroon [8]. Fifty-three villages were visited and an overall prevalence of 5.8% was found [8]. These regions included localities where four cross-species transfers of simian immunodeficiency virus (SIV) from non-human primates to humans occurred [9,10,11]. One of these transfers gave rise to HIV-1 group M (HIV-1M), which is today responsible for the global HIV pandemic. It is plausible that after sustaining human-to-human transmission, the HIV-1M progenitor spread throughout Cameroon’s equatorial rainforest region and sparked localised epidemics [12]. This suggests that these regions host some of the oldest and most mature HIV-1M epidemics.

In 2016, Cameroon began implementing the “Universal Test and Treat” (UTT) [13] policy to advance towards the UNAIDS target of 95-95-95 by 2025, whereby by 2025, we should have diagnosed, treated, and suppressed viral loads in 95% of people living with HIV. As a result, the number of people living with HIV (LWH) on antiretroviral treatment (ART) increased from 168,431 (27.1% ART coverage) in 2015 to 350,818 (70.6% ART coverage) in 2020 [14]. The most recent Demographic and Health Survey (DHS) conducted in 2018 indicated an overall HIV prevalence of 2.8% [15], a decrease from 5.3% in 2014 [16].

Ref. [17] reviewed the three DHSs organised in the country in 2018, 2014, and 2011 and found that there were some persistent disparities in prevalence by age, regions, and gender among individuals infected with HIV. For instance, women showed higher infection rates than men (3.5% versus 1.9%), with adults aged 35–39 years being the most affected. Regional differences ranged from 5.8% in the South to 1% in the Far North region [17]. In addition, their analyses also indicated that the decline in HIV prevalence in Cameroon over the study period was associated with a reduction in risky sexual behaviours, including unprotected sex and multiple partners, and limited HIV testing uptake. Notably, the proportion of men and women receiving an HIV test and result increased by more than 30% [17]. Furthermore, among the 20 locations that were identified as HIV hotspot clusters during the 2018 survey, 13 were rural or semi-rural localities [18]. Therefore, understanding the socio-cultural factors behind higher rural HIV infection is critical for the development of targeted interventions.

The aim of this study was to estimate HIV seroprevalence and identify associated socio-demographic factors in remote, rural communities of Cameroon’s equatorial rainforest regions. We also sought to compare prevalence across sex, age, and region, with particular emphasis on adults aged 50 years and older, in order to address the limitations of conventional surveillance systems that primarily focus on the 15–49-year-old age group.

## 2. Materials and Methods

### 2.1. Villages and Study Population

We conducted a health campaign in 2021 and 2022 in remote communities of four administrative regions within Cameroon’s equatorial rainforest (Figure 1). Villages were selected based on the known acceptability of campaigns to their inhabitants and the availability of local staff. The campaign addressed general health issues (STD, hypertension, malaria, and food safety) and included voluntary HIV testing. Inclusion criteria comprised individuals aged 15 years or older who wished to know their HIV status and were without obvious comorbidities, referred to as visible or known health conditions at the time of enrolment, such as active tuberculosis, severe chronic illnesses (e.g., advanced cardiac or respiratory failure), or any condition requiring urgent medical care; the exclusion criteria involved any individuals with obvious comorbidities, those who were unwilling to know their HIV status, and/or those who were under the age of 15 years. The campaign started with door-to-door awareness by local health workers, followed by a mobile clinic in a public village space over two days. Demographic and behavioural information was collected via face-to-face interviews using a structured questionnaire. Pre- and post-test counselling was performed per Cameroonian national guidelines [13]. Blood samples were collected after pre-counselling for HIV testing. The survey was conducted in conjunction with local public health personnel who also ensured that individuals diagnosed with HIV were referred to the nearest healthcare centre for access to ART according to national guidelines and procedures [13]. Five ml of blood was drawn from all participants for HIV testing.

### 2.2. HIV Testing

Blood samples were transported to a specialised HIV testing laboratory under optimal cold chain conditions to maintain test reliability [19]. They were then tested for HIV antibodies. Four different tests were used to diagnose the HIV infection: two ELISA tests, the HUMAN^®^HIVAg/Ab (HUMAN Diagnostics Worldwide, Wiesbaden, Germany), and the Murex HIV-1.2.0 ELISA (DiaSorin, Dartford, UK), along with two rapid tests, the Determine™ HIV Early Detect (Abbott Diagnostic Medical Co. Ltd., Matsudo, Japan) and the MULTISURE^®^ HIV Rapid Test (MP Diagnostics, Asia Pacific, Singapore). We adopted the WHO three tests strategy as the reported national HIV prevalence in the country is below 5% [20]. The testing algorithm is summarised in Figure 2. The HUMAN^®^HIVAg/Ab was used to pre-screen samples, with non-reactive results considered HIV-negative. Reactive samples (potentially HIV-positive) were then retested using either the Murex HIV-1.2.0 ELISA or the Determine™ HIV Early Detect. Plasma samples reactive to these tests were finally tested using the MULTISURE^®^ HIV Rapid Test and reactive samples were classified as HIV-positive. On the other hand, discordant samples (reactive to one of these three tests) were repeated and if the same results were obtained, they were categorised as “indeterminate” and were excluded from further analyses (Figure 2). Participants with such a result were requested to perform another test in the nearest health centre after one month while abstaining from any risky behaviour.

### 2.3. Statistical Analyses

Data were analysed using Stata version 19 [21]. Frequencies and percentages (%) were used to describe qualitative variables. Pearson’s chi-square test was used to establish relationships between qualitative variables. Since this was a cross-sectional study and we aimed to estimate prevalence ratios (PRs) directly, the modified Poisson regression method may have been appropriate, because PRs are more interpretable than odd ratios (ORs) in cross-sectional studies (Barros and Hirakata [22]). However, Ibáñez-Pinilla et al. [23] recently showed that the negative log-binomial model with robust variance outperforms the modified Poisson regression. Since different villages were sampled, we accounted for intra-cluster correlation by using the negative log-binomial model with generalised estimating equations (GEEs) and obtained unadjusted PRs for univariable (bivariate) analysis and adjusted prevalence ratios (aPRs) for multivariate analysis with 95% confidence interval (CI). Variables used for adjustment were those with *p*-values less than 0.05 at univariate analysis; these included age group (*p* = 0.017), sex (*p* = 0.005), region (*p* < 0.001), and marital status (*p* < 0.001). Finally, variables with *p*-values less than 0.05 were considered statistically significant. The goodness of fit was measured with Quasi Likelihood under Independence Model Criterion (QIC).

### 2.4. Ethics Considerations

The study was approved on the 19 December 2022 by the Cameroon Ministry of Public Health through its National Ethics Committee (N° 2022/12/1510/CE/CNERSH/SP). Written informed consent was distributed to all participants and signed. HIV-positive individuals received counselling and referral to the nearest ART centre.

## 3. Results

### 3.1. Description of Serosurveys

During January 2021 and September 2022, we visited 46 villages and enrolled 5631 participants from four administrative regions of the equatorial rainforest areas of Cameroon: the Centre, East, Littoral and South (Figure 1). A total of 55 participants were excluded from the study due to indeterminate HIV results (defined in the Methods Section) leaving a number of 5576 study participants. In the Centre region, 9 villages were visited and 829 individuals enrolled (15% of the total participants); in the Littoral region, 3 villages and 859 individuals (15%); in the South, 23 villages and 1905 individuals (34%); and in the East, 11 villages and 1983 individuals (36%).

### 3.2. Characteristics of Study Participants

Among the participants who responded to the questionnaire, the majority (60.6%) of them were male. Participant ages varied from 15 to 101 years with median age 31 (IQR: 22–42) years. The age group 15–19 years was the most represented, accounting for 18.7%. The majority (90%) of the study participants had attended at least a primary school and 58% of them were in a live-in relationship (officially married or not).

### 3.3. HIV Prevalence in Remote Communities of Cameroon

The overall HIV prevalence among our study participants was 3.6% (95% CI: 3.2–4.1%) with notable variation among the four administrative regions (*p* < 0.001). Compared to the Centre region, which had the lowest prevalence at 2.3%, the East (2.8%) and the Littoral (3.3%) regions had a similar prevalence (*p* = 0.43 and 0.23, respectively). However, the South (5.2%) had a significantly higher prevalence (*p* < 0.001) (Table 1). Among the visited villages, the prevalence ranged from 0.5% in Ngoumou in the Centre region to 14.0% in Mekoto in the South region.

### 3.4. Bivariate Analyses

Table 1 also details the prevalence according to gender, age groups, marital status, and level of education. As shown in this table, women were 1.5 times more infected than men [PR (95% CI): 1.48(1.13–1.94), *p* = 0.005].

Marital status was categorised as single, married, divorced/separated, or living together. HIV prevalence was similar among single individuals and those in stable unions (both married or living together) but significantly higher in individuals who were divorced or separated (*p* < 0.001). Individuals with primary education were 1.86 times more likely to be infected than those with no education [PR (95% CI): 1.86(0.68–5.10)], but this was not significant (*p* = 0.23). Similarly, the prevalence among those with secondary or higher education was similar to those with no education (3.1% vs. 2.1%).

Analysis of age groups showed the lowest prevalence in the 15–19 years group (1.8%). Apart from the 30–34 years group, all older cohorts had a significantly higher HIV prevalence compared to this baseline. In addition, we observed a striking contrast between genders. Peak prevalence was observed among women aged 35–44 years (over 6%) and 55 years and older (over 5%), while among men, the most affected group were those aged 45–54 years (over 5%) (Table 1 and Figure 3). Furthermore, when participants were grouped into standard epidemiological strata of 15–49 years (generally used in studies or surveys to determine the prevalence in Cameroon) vs. ≥50 years, prevalence was significantly higher in the latter (4.9%) vs. 3.4% in the former (*p* = 0.025; Table 1).

### 3.5. Multivariate Analyses

Table 2 presents the results from the multivariate analysis. After adjustment, the factors detailed below remained significantly associated with HIV seropositivity. The multivariate analysis revealed a higher HIV prevalence ratio among individuals aged 50–54 years compared to those aged 15–19 years (aPR = 2.76, 95% CI: [1.24–6.15], *p* = 0.013). Women had higher odds (aPR = 1.53, 95% CI: [1.12–2.08], *p* = 0.007) than men. Compared to the Centre region, risk was significantly higher in the South (aPR = 1.82, 95% CI: [1.04–3.18], *p* = 0.035).

Marital status was significantly associated with HIV status (*p* = 0.019). In fact, divorced/separated/widowed individuals had increased risk (aPR = 1.70, 95% CI: [0.80–3.58], *p* = 0.165) compared to single individuals, although this was not significant (0.165).

## 4. Discussion

The present study shows differences in HIV infection and associated risk factors in remote communities of four administrative regions of the equatorial rainforest of Cameroon, a country where HIV-1 group M (HIV-1M) has likely been circulating since the start of the global HIV-1 pandemic and with one of the highest HIV-1M genetic diversities [24]. We observed significant regional differences, with the lowest prevalence found in the Centre region and the highest in the South. Furthermore, women were significantly more likely to be infected than men, and HIV prevalence was highest among divorced/separated individuals compared to participants with single status. Additionally, our data show that adults (aged 35 years and above) have markedly higher HIV prevalence than adolescents and young adults. Notably, HIV prevalence in the population aged 50 years and older was 4.9%, significantly higher than the 3.4% reported among individuals aged 15 to 49 years, a group typically used as the standard denominator in national HIV surveys in Cameroon.

The overall prevalence of 3.4% among individuals aged between 15 and 49 years (at 95% CI: 2.9–3.9%) was higher than that indicated by the most recently published study from 2018 that focused on urban and semi-urban locations and found that the prevalence of HIV infection was 2.7% [25] and 2.8% [15]. One key observation should be mentioned here: rural communities in the South region experienced the highest HIV prevalence (5.2%), as was the case with individuals living in urban areas of this same region (6.9%) [25]. In addition, we also found a similar prevalence of HIV infection in remote localities of the South in 2000 (4.5%) [8] and in 2012/2013 (5.5%) [26]. This observation seems to indicate that HIV prevalence in the South region of Cameroon has been stable over time at around 5%. The South region of Cameroon is well connected to other parts of the country, and more importantly, to many neighbouring countries (Gabon, Equatorial Guinea, and Republic of Congo). This high connectivity certainly lowers barriers to HIV spread by increasing mixing of diverse populations. In addition, the difference might also be attributed to the availability and quality of health services for the management of HIV/AIDS across the country. Specifically, when the national AIDS control committee assessed factors like the accessibility of HIV testing and counselling, the intensity of efforts directed towards the prevention of mother-to-child transmissions, care and support of people living with HIV, and the provision of laboratory and medication management throughout health facilities of Cameroon, they found high discrepancies among health facilities with less resources [27]. It is also worth noting that many of the visited communities are located along the road network used by truck drivers, especially in the East, Littoral and South regions, and many studies have described this specific group of individuals as a high-risk population for HIV infection [19,28].

The main finding of this study is that HIV circulates mainly among adults and older individuals, emphasising the importance of including older adults in surveillance studies. Two peaks in prevalence were observed among women: those aged 35–45 years and those aged 55–64 years and above. Among men, the peak occurred in the 45–54 years group, which suggests a mature and possibly long-standing epidemic. In contrast, the most recent Demographic and Health Survey (DHS) conducted in 2018 in urban and semi-urban areas found that adults aged 35–39 years were the most affected group [15]. Three main hypotheses may explain the relatively high prevalence in older individuals observed in our cohort. First, sample bias may play a role: fewer individuals (ranging from 2% to 9%) were enrolled in the 40–65 years and over age groups compared with those aged 15–39 years (12–19%), and imprecise estimates are more likely with smaller sample sizes [29]. Second, the expansion of ART may have increased the life expectancy of PLWH, allowing those infected as adults to survive into older age. Third, demographic shifts such as rural exodus may have led to a higher proportion of older adults remaining in rural areas, as younger people migrate to cities for employment. Nevertheless, assuming that older women are less sexually active as their younger counterparts, one should worry about older men aged between 45 and 54 years who can be a source of new infections in the communities. We acknowledge that the selection of study participants could have been biassed toward health-motivated individuals and this could have inflated estimates of HIV prevalence; nevertheless, we also think that because this group of health-motivated individuals are mixed with other healthy individuals in the community, there is not expected to be a significant HIV prevalence difference between these individuals and those in the general population.

Finally, this study has several strengths and limitations. Strengths include the large sample size, the use of a rigorous diagnostic strategy aligned with WHO guidelines [20], and the focus on rural and remote areas of Cameroon that are often underrepresented in national surveys. However, this study also has several limitations, including the non-random sampling design, volunteer-related selection bias, small sample sizes in certain subgroups, and the lack of detailed analyses of sexual behaviours. In addition, clinical data (viral load and CD4 count) and ART coverage were not available to distinguish prevalent vs. undiagnosed cases.

## 5. Conclusions

This study demonstrates that HIV remains a significant health concern in remote, rural communities of Cameroon’s equatorial regions, with patterns distinct from those in urban areas. Older adults, particularly those aged 35 years and above, and women bear a disproportionate burden of infection. Regional disparities were evident, with the South region consistently showing the highest prevalence, suggesting persistent local drivers of transmission. Divorce and widowhood were also associated with elevated risk, underscoring the role of social vulnerability. The concentration of cases among older age groups, likely influenced by ART-driven survival and demographic shifts, challenges the conventional focus on the 15–49 years age bracket in surveillance. Despite some limitations on sampling bias and size, our data underscore the need to expand monitoring to include older adults, and tailoring prevention and care strategies to high-prevalence regions and mobile populations are critical steps. Addressing these gaps is essential to curb ongoing transmission and achieve equitable HIV control in rural Cameroon.

## Figures and Tables

**Figure 1 tropicalmed-10-00334-f001:**
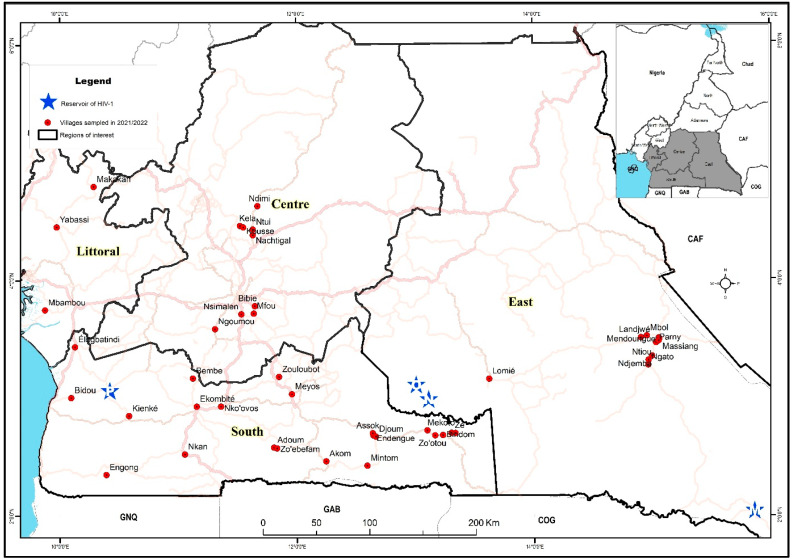
Map showing the geographical location of the study sampling sites in the four regions of the equatorial zone of Cameroon. In red are the villages sampled in 2021/2022 and in blue are the sites where HIV-1 groups M, N, O, and P were identified.

**Figure 2 tropicalmed-10-00334-f002:**
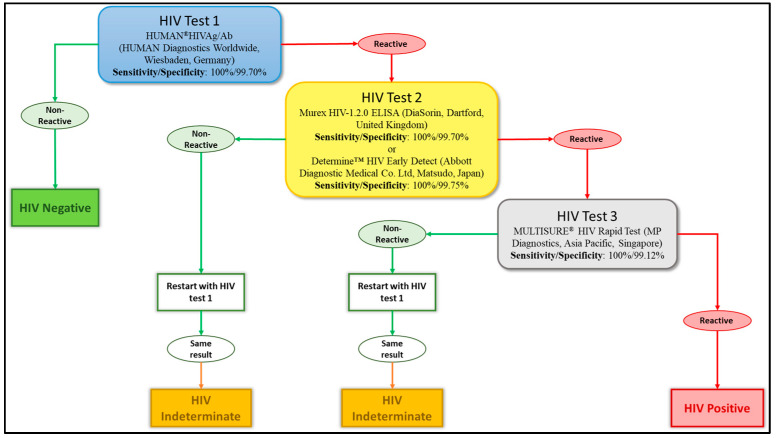
Diagram detailing the national algorithm for the HIV testing strategy. This algorithm recommends performing three successive tests. Each level offers the possibility of two different results: reactive or non-reactive. When all three tests are reactive for the same sample, an HIV-positive diagnosis is made. If the sample is non-reactive in one of the tests after being declared reactive in the first, the result is indeterminate. Lastly, if the first test does not react, the sample is declared HIV-negative. Sensitivity and specificity were provided by the manufacturer of the different tests.

**Figure 3 tropicalmed-10-00334-f003:**
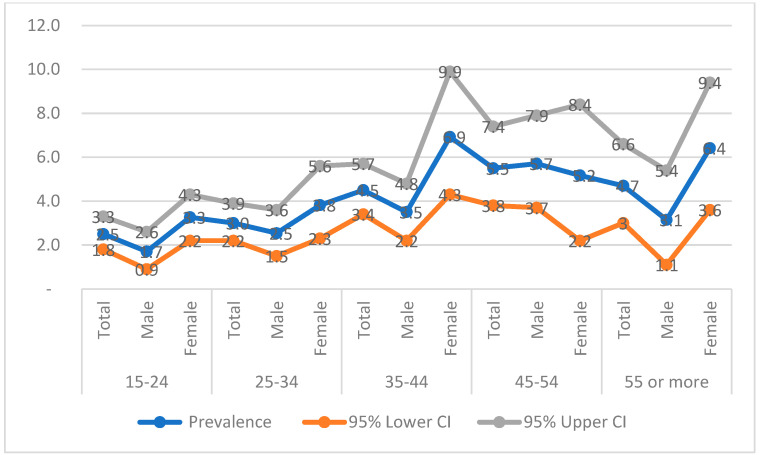
Trend curves for HIV prevalence (%) by sex and age (with 95% confidence interval (CI)), associated with the overall trend generated from the prevalence rates obtained for each desired age group (Years) of patients sampled at the sites visited as part of the study.

**Table 1 tropicalmed-10-00334-t001:** Seroprevalence of HIV and associated socio-demographic factors (univariate analysis).

Characteristics	Negative n (%)	Positive n (%)	Total	PR (95% CI)	*p*-Value
Age (Years, *p* = 0.017, Quasi Likelihood under Independence Model Criterion (QIC) = 1265.76)					
15–19	1022(98.2)	19(1.8)	1041	1	
20–24	703(96.4)	26(3.6)	729	1.95(1.09–3.50)	0.025
25–29	676(96.7)	23(3.3)	699	1.80(1.00–3.28)	0.05
30–34	719(97.3)	20(2.7)	739	1.48(0.80–2.76)	0.21
35–39	661(95.4)	32(4.6)	693	2.53(1.45–4.43)	0.001
40–44	477(95.6)	22(4.4)	499	2.42(1.32–4.42)	0.004
45–49	309(94.5)	18(5.5)	327	3.02(1.60–5.68)	<0.001
50–54	274(94.5)	16(5.5)	290	3.02(1.58–5.80)	<0.001
55–59	164(94.8)	9(5.2)	173	2.85(1.31–6.20)	0.008
60–64	129(96.3)	5(3.7)	134	2.04(0.78–5.38)	0.15
65 or more	240(95.2)	12(4.8)	252	2.61(1.28–5.30)	0.008
**Age (Years, *p* = 0.025, Quasi Likelihood under Independence Model Criterion (QIC) = 1204.63)**					
15–49	4567(96.6)	160(3.4)	4727	1	
50 or more	807(95.1)	42(4.9)	849	1.46(1.05–2.04)	0.025
**Sex (*p* = 0.005, Quasi Likelihood under Independence Model Criterion (QIC) = 1191.79)**					
Male	3261(97.0)	102(3.0)	3363	1	
Female	2085(95.5)	98(4.5)	2183	1.48(1.13–1.94)	0.005
**Region (*p* < 0.001, Quasi Likelihood under Independence Model Criterion (QIC) = 1207.65)**					
Centre	810(97.7)	19(2.3)	829	1	
East	1927(97.2)	56(2.8)	1983	1.23(0.74–2.06)	0.43
Littoral	831(96.7)	28(3.3)	859	1.42(0.80–2.53)	0.23
South	1806(94.8)	99(5.2)	1905	2.27(1.40–3.68)	<0.001
**Marital Status (*p* < 0.001, Quasi Likelihood under Independence Model Criterion (QIC) = 1003.47)**					
Single	1829(96.8)	60(3.2)	1889	1	
Married	1423(96.9)	46(3.1)	1469	0.99(0.68–1.44)	0.94
Divorced/Separated /Widow(er)	115(89.8)	13(10.2)	128	3.20(1.80–5.67)	<0.001
Living together	1278(96.7)	43(3.3)	1321	1.02(0.70–1.51)	0.90
**Education (*p* = 0.5, Quasi Likelihood under Independence Model Criterion (QIC) = 843.54)**					
No education	187(97.9)	4(2.1)	191	1	
Primary	1326(96.4)	49(3.6)	1375	1.86(0.68–5.10)	0.23
Secondary or higher	2497(96.9)	80(3.1)	2577	1.62(0.60–4.38)	0.34

**Table 2 tropicalmed-10-00334-t002:** Multivariate analysis for HIV-associated socio-demographic factors using negative log-binomial model with generalised estimating equations (GEEs).

Characteristics	aPR (95% CI)	a*P*-Value
**Test of model main effects: Age (*p* = 0.17, Wald chi-square = 14.04)**		
15–19	1	
20–24	1.61(0.85–3.05)	0.14
25–29	1.64(0.84–3.23)	0.15
30–34	1.42(0.69–2.93)	0.34
35–39	2.15(1.10–4.20)	0.02
40–44	2.04(0.97–4.31)	0.06
45–49	3.11(1.46–6.64)	0.003
50–54	2.76(1.24–6.15)	0.013
55–59	2.98(1.23–7.20)	0.015
60–64	2.29(0.82–6.40)	0.11
65 or more	1.82(0.69–4.80)	0.22
**Test of model main effects: Sex (*p* = 0.007, Wald chi-square = 7.24)**		
Male	1	
Female	1.53(1.12–2.08)	0.007
**Test of model main effects: Region (*p* = 0.167, Wald chi-square = 5.07)**		
Centre	1	
East	1.38(0.79–2.41)	0.25
Littoral	1.51(0.81–2.81)	0.196
South	1.82(1.04–3.18)	0.035
**Test of model main effects: Marital status (*p* = 0.019, Wald chi-square = 10.00)**		
Single	1	
Married	0.65(0.41–1.02)	0.059
Divorced/Separated/Widow(er)	1.70(0.80–3.58)	0.165
Living together	0.82(0.53–1.26)	0.367
**Goodness of fit: Quasi Likelihood under Independence Model Criterion (QIC) = 1109.62**		

## Data Availability

The original contributions presented in this study are included in the article. Further inquiries can be directed to the corresponding author.

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
