# Peer review of "Epidemiology of HIV in Remote Equatorial Regions of Cameroon: High Prevalence in Older Adults and Regional Disparities"

_tropicalmed, 2025, doi:10.3390/tropicalmed10120334_

Round 1
Reviewer 1 Report
Comments and Suggestions for Authors
The topic is relevant to healthcare providers, policymakers, program managers, and the general public. The title reflects the content of the manuscript. The abstract is an adequate summary of the manuscript. The introduction orients the readers to the topic and identifies a literature gap. However, the aim of the study is not clearly stated at the end of the introduction. The materials and methods section is clearly described to allow for the replication of the study. The results are clearly presented with the help of tables and figures. The authors compare their findings with previous literature in the discussion section. However, the authors did not discuss the strengths and limitations of their study. The conclusion is based on the study’s findings.
MINOR REVISIONS
Introduction
- In lines 73-75, the authors state, ‘In addition, their analyses also indicated that the decline in HIV prevalence in Cameroon over the study period was associated with a reduction in sexual behaviors.’ The authors should clarify the sexual behaviors they are referring to. Is it risky sexual behaviors?
Results
- In line 172, the authors state, ‘The overall HIV prevalence among our study participants was 3.6% …’ The authors should provide the 95% confidence interval of the overall prevalence.
- In lines 200-201, the authors state, ‘….emphasizing the importance of including older adults in surveillance studies.’ This statement must be moved to the discussion section.
- In lines 257-259, the authors state, ‘The overall prevalence among the individuals aged between 15 to 49 years at 3.4% was similar to the most recent data, published in 2018 and organized in urban and semi-258 urban locations, that indicated a rate of HIV infection of 2.7%[22] and 2.8%[14].’ Without stating the 95% confidence interval in the results section, the authors cannot say the overall prevalence is similar to the other 2.
Discussion
- The strengths and limitations of the study should be included at the end of the discussion section.
Author Response
The topic is relevant to healthcare providers, policymakers, program managers, and the general public. The title reflects the content of the manuscript. The abstract is an adequate summary of the manuscript. The introduction orients the readers to the topic and identifies a literature gap. However, the aim of the study is not clearly stated at the end of the introduction. The materials and methods section is clearly described to allow for the replication of the study. The results are clearly presented with the help of tables and figures. The authors compare their findings with previous literature in the discussion section. However, the authors did not discuss the strengths and limitations of their study. The conclusion is based on the study’s findings.
Answers: Thank you. Regarding the aim, we have revised to make it clearer as outlined below and in the text:
The aim of this study was to estimate HIV seroprevalence and identify associated sociodemographic factors in remote rural communities of Cameroon’s equatorial rainforest regions. We also sought to compare prevalence across sex, age, and region, with particular emphasis on adults aged 50 years and older, in order to address the limitations of conventional surveillance systems that primarily focus on the 15-49 year old age group
In addition, we have included the strengths and limitations of the study in the discussion as follow:
Finally, this study has several strengths and limitations. Strengths include the large sample size, the use of a rigorous diagnostic strategy aligned with WHO guidelines[20], and the focus on rural and remote areas of Cameroon that are often underrepresented in national surveys.
However this study also has several limitations including the non-random sampling design, volunteer-related selection bias, small sample sizes in certain subgroups, and the lack of detailed analyses of sexual behaviors. In addition, clinical data (viral load, CD4 count) and ART coverage were not available to distinguish prevalent vs. undiagnosed cases.
MINOR REVISIONS
Introduction
1. In lines 73-75, the authors state, ‘In addition, their analyses also indicated that the decline in HIV prevalence in Cameroon over the study period was associated with a reduction in sexual behaviors.’ The authors should clarify the sexual behaviors they are referring to. Is it risky sexual behaviors?
Answer: This is a good point. We were referring to risky sexual behaviours including multiple partners, unprotected sex, limited HIV testing uptake. The text has been revised.
Results
2. In line 172, the authors state, ‘The overall HIV prevalence among our study participants was 3.6% …’ The authors should provide the 95% confidence interval of the overall prevalence.
Answer: We welcome this suggestion and this was added as follows: ‘The overall HIV prevalence among our study participants was 3.6% (95%CI: 3.2%-4.1%)’
3. In lines 200-201, the authors state, ‘….emphasizing the importance of including older adults in surveillance studies.’ This statement must be moved to the discussion section.
Answer: The statement: “emphasizing the importance of including older adults in surveillance studies” has been moved to the discussion section.
4. In lines 257-259, the authors state, ‘The overall prevalence among the individuals aged between 15 to 49 years at 3.4% was similar to the most recent data, published in 2018 and organized in urban and semi-urban locations, that indicated a rate of HIV infection of 2.7%[22] and 2.8%[14].’ Without stating the 95% confidence interval in the results section, the authors cannot say the overall prevalence is similar to the other 2.
Answer: after adding the 95% confidence interval, we adjusted the paragraph as below:
“The overall prevalence of 3.4% among individuals aged between 15 to 49 years at (95%CI: 2.9%-3.9%), was higher than that indicated by the most recent previously published study from 2018 that focused on urban and semi-urban locations and found the prevalence of HIV infection was 2.7%[25] and 2.8%[15]
Discussion
5. The strengths and limitations of the study should be included at the end of the discussion section.
Answers: Thank you. We have included the strengths and limitations of the study as recommended.
Reviewer 2 Report
Comments and Suggestions for Authors
Thank you for inviting me to review this this manuscript, which reports findings from a large-scale, community-based cross-sectional survey of HIV seroprevalence in remote rural areas of Cameroon. The authors examine HIV prevalence patterns across age, sex, and region, with particular emphasis on older adults and regional disparities. The study is framed as a contribution to the growing recognition that rural and older populations are often underrepresented in routine surveillance efforts. The study has important strengths: the real-world nature of the data, and mode of collection through an integrated testing campaign adds practical value, and the diagnostic strategy employed is rigorous and aligned with WHO guidelines.
Nonetheless, the manuscript has important conceptual, methodological, analytical, and reporting shortcomings that must be addressed before it can be considered for publication. I elaborate on these below, including options for authors to strengthen each section of the manuscript
Introduction
The introduction is underdeveloped and can be better contextualized. It begins with broad global statistics, some of which are outdated, and fails to establish a clear rationale for the current study. For example, the citation of UNAIDS figures stating 1.3 million new infections in 2024 and 44% in women and girls does not directly connect to Cameroon’s national or regional context. There is no discussion of Cameroon’s HIV burden, the urban-rural divide in surveillance coverage, or the unique dynamics of the equatorial forest regions. Most concerning is the absence of a clearly articulated research gap: the text does not explain why this study was necessary or what it adds beyond existing national surveys. A stronger introduction would begin with a brief overview of Cameroon’s current HIV epidemiology (e.g., ~3.8% national adult prevalence per UNAIDS 2023), then narrow the focus to rural disparities, the specific challenges of remote equatorial regions (e.g., mobility, poor access to ART), and the systematic exclusion of older adults from many surveillance efforts. Prior work on regional variation or older adult vulnerability should be cited to frame the hypotheses. The rationale for using a community campaign to capture underserved populations should be clarified. Overall, the introduction should be expanded and include at least 10–15 relevant citations to support claims.
Method: The major flaw of the paper is in the Methods, which seems fragmented and omits several key components needed to assess the reliability of the findings. Firstly, the sampling approach is based on convenience sampling through health campaigns, but the authors do not specify how villages were selected, what proportion of the target population participated, or what the response rate was. This raises serious concerns about selection bias, particularly volunteer bias, which likely skews the sample toward health-motivated individuals and could inflate estimates of HIV prevalence in older adults. In the same vein, eligibility criteria are vague. For instance, “obvious comorbidities” is not operationalized, and requiring “willingness to be tested” introduces self-selection effects. The implications of these choices for bias are not acknowledged.
The diagnostic algorithm used is a relative strength of the paper. It follows WHO guidelines for low-prevalence settings (<5%) and includes a well-described series of rapid and confirmatory assays, ensuring internal validity of HIV test results. However, the logistics of blood collection—5 mL samples in remote and hot environments are not sufficiently described. There is no information on sample transport, storage conditions, or cold chain management, which are crucial in determining test reliability in field conditions. Authors could have referenced other papers where the survey procedures have been completely documented, or put the details as supplementary material.
Perhaps the most important shortcoming of the methodology is the statistical analysis. The outcome of interest (HIV status) is a binary outcome (yes/no), yet the authors rely on Poisson and negative binomial regressions, which are designed for count data. While modified Poisson regression can be validly used for binary outcomes to estimate prevalence ratios (as per Barros and Hirakata, 2003), the authors do not explain their rationale or address critical assumptions. Their subsequent shift to negative binomial regression due to “overdispersion” is methodologically unjustified. Overdispersion in binary data is conceptually different from that in count data and typically points to issues like clustering or model misspecification. There is no evidence presented (e.g., Pearson chi-square or deviance/df ratios) to support this decision, nor is the dispersion parameter reported.
Using negative binomial regression for a binary outcome risks biased estimates and incorrect standard errors. It introduces assumptions (such as gamma-distributed heterogeneity) that are not applicable to binary outcomes. Standard practice in cross-sectional HIV seroprevalence studies is to use logistic regression or, where prevalence is low, modified Poisson regression with robust variance estimators. Log-binomial regression is another valid method but may face convergence issues. None of these alternatives are discussed. There is also no assessment of multicollinearity, interaction terms (e.g., age by sex or region), or model diagnostics such as AIC, BIC, or pseudo-R². This analytical approach undermines the validity of the reported adjusted prevalence ratios (e.g., aPR = 2.27 for the South region; aPR = 1.48 for females).
In addition, the authors ignore clustering in the data. With 46 villages sampled, standard errors are likely deflated due to intra-cluster correlation. This could explain the perceived “overdispersion” and should have been addressed using generalized estimating equations (GEE) or multilevel modeling approaches (e.g., mixed-effects logistic regression). The lack of adjustment for clustering is a serious flaw.
Behavioral data reportedly collected in the questionnaire, such as sexual history, condom use, or number of partners, are not analyzed or even described in the results. This is a missed opportunity, particularly given the manuscript’s emphasis on “epidemiology” and “risk factors.” The authors should either include analyses of these variables or clarify their absence.
Results
The results section is generally well organized and includes helpful tables and figures, but several inconsistencies and errors undermine their clarity. Table 1 presents demographic characteristics of participants, but the sex distribution (60.6% male) is unusual for HIV testing surveys, which typically oversample women. The median age of 31 years is plausible, but the interquartile range (22–42) suggests a youth skew, which appears at odds with the manuscript’s emphasis on older adult prevalence.
Figure 3 clearly shows the age-sex breakdown of HIV prevalence and successfully highlights peaks among women aged 35–44 and 55+, and men aged 45–54. However, the sample sizes for older age groups are small (often <10% per band), leading to potential instability in prevalence estimates. Confidence intervals should be added to illustrate this.
Table 2 presents adjusted prevalence ratios but does not clearly distinguish between crude and adjusted models. The p-values are inconsistently formatted (e.g., use of commas for decimals) and some categories are poorly labeled or duplicated (e.g., “50 or more” appears twice). There are also typographical errors throughout. Importantly, there is no model fit information reported (e.g., goodness-of-fit tests, AIC/BIC), nor are diagnostics provided. The authors also exclude 55 individuals with indeterminate results (~1%), but do not test whether their exclusion affects the findings.
Discussion
The discussion is the manuscript’s strongest section. It thoughtfully connects findings to the broader literature, such as the consistently elevated prevalence in the South region (5.2%), which aligns with previous studies suggesting a stable, long-standing epidemic in this area. The linkage of male vulnerability to mobility and trade routes is also well reasoned. The authors correctly critique the standard practice of excluding older adults (≥50) from surveillance and propose important hypotheses, such as survival bias among ART users or rural exodus returning older people to their home villages.
That said, the discussion overreaches in several places. It refers to a “mature and long-standing epidemic” without supporting incidence or temporal data. It attributes the education-HIV association (more educated individuals having higher prevalence) to vulnerability without considering that this may reflect reverse causality or residual confounding. Comparisons to urban DHS data are problematic because the current sample is rural and non-random. The discussion of HIV-1 subtype diversity is tangential and underexplored. Limitations are only partially acknowledged: selection bias is noted, but not quantified; the omission of behavioral variables and the use of flawed regression models are ignored. There is no mention of ART coverage or viral load status, both of which would help distinguish prevalent vs. undiagnosed cases. The discussion would benefit from a clearer acknowledgment of these limitations.
Conclusions
The conclusions are concise and correctly call for surveillance systems to better capture older adults and underserved regions. However, they overstate the evidence by referring to “persistent drivers” without having measured any such mediating factors. The conclusions should align more closely with the limitations of the study, particularly the non-probability sampling and flawed analysis. Calls for inclusion in UNAIDS targets are valuable but should be framed with appropriate caveats about data quality.
Figures and Tables
Figure 2 (diagnostic algorithm) is well-designed but should include test sensitivity/specificity. Figure 3 effectively shows prevalence curves but lacks confidence intervals. Tables 1–2 need to be cleaned for typos, formatting consistency, and inclusion of 95% confidence intervals.
Major revisions are necessary, particularly in the statistical approach, contextual framing, and interpretation of results. If addressed rigorously, the study could make a meaningful contribution to rural HIV epidemiology and policy.
Comments on the Quality of English LanguageThorough language editing should be performed
Author Response
Thank you for inviting me to review this this manuscript, which reports findings from a large-scale, community-based cross-sectional survey of HIV seroprevalence in remote rural areas of Cameroon. The authors examine HIV prevalence patterns across age, sex, and region, with particular emphasis on older adults and regional disparities. The study is framed as a contribution to the growing recognition that rural and older populations are often underrepresented in routine surveillance efforts. The study has important strengths: the real-world nature of the data, and mode of collection through an integrated testing campaign adds practical value, and the diagnostic strategy employed is rigorous and aligned with WHO guidelines.
Nonetheless, the manuscript has important conceptual, methodological, analytical, and reporting shortcomings that must be addressed before it can be considered for publication. I elaborate on these below, including options for authors to strengthen each section of the manuscript
Introduction
The introduction is underdeveloped and can be better contextualized. It begins with broad global statistics, some of which are outdated, and fails to establish a clear rationale for the current study. For example, the citation of UNAIDS figures stating 1.3 million new infections in 2024 and 44% in women and girls does not directly connect to Cameroon’s national or regional context. There is no discussion of Cameroon’s HIV burden, the urban-rural divide in surveillance coverage, or the unique dynamics of the equatorial forest regions. Most concerning is the absence of a clearly articulated research gap: the text does not explain why this study was necessary or what it adds beyond existing national surveys. A stronger introduction would begin with a brief overview of Cameroon’s current HIV epidemiology (e.g., ~3.8% national adult prevalence per UNAIDS 2023), then narrow the focus to rural disparities, the specific challenges of remote equatorial regions (e.g., mobility, poor access to ART), and the systematic exclusion of older adults from many surveillance efforts. Prior work on regional variation or older adult vulnerability should be cited to frame the hypotheses. The rationale for using a community campaign to capture underserved populations should be clarified. Overall, the introduction should be expanded and include at least 10–15 relevant citations to support claims.
Answer: We appreciate this suggestion.
As mentioned in our manuscript, published data on HIV prevalence in Cameroon are scarce. Recent data on this topic have been based on demographic and health surveys. These were referenced in the manuscript. However we have expanded the introduction with the following paragraph:
Cameroon has an estimated national adult HIV prevalence between 2.5% and 3.8% (UNAIDS, 2020), with marked disparities across regions and sexes. However, existing surveillance systems primarily focus on urban areas and the 15–49 years age group, leaving limited data on rural populations and adults aged 50 years and above. In this context, the present study aimed to estimate HIV seroprevalence and identify associated sociodemographic factors in remote rural communities of Cameroon’s equatorial rainforest regions, in order to address these gaps.
Method: The major flaw of the paper is in the Methods, which seems fragmented and omits several key components needed to assess the reliability of the findings. Firstly, the sampling approach is based on convenience sampling through health campaigns, but the authors do not specify how villages were selected, what proportion of the target population participated, or what the response rate was. This raises serious concerns about selection bias, particularly volunteer bias, which likely skews the sample toward health-motivated individuals and could inflate estimates of HIV prevalence in older adults. In the same vein, eligibility criteria are vague. For instance, “obvious comorbidities” is not operationalized, and requiring “willingness to be tested” introduces self-selection effects. The implications of these choices for bias are not acknowledged.
Answer: This is a good point. Villages were selected based on the known acceptability of campaigns to their inhabitants and the availability of local staff. This was also added in the text.
In addition, in our study, “obvious comorbidities” referred to visible or known health conditions at the time of enrolment, such as active tuberculosis, severe chronic illnesses (e.g., advanced cardiac or respiratory failure), or any condition requiring urgent medical care. These criteria were intended to avoid including individuals who needed specialized follow-up that could not be provided within the framework of the community campaign. We have modified the manuscript accordantly.
Finally, we discussed the sampling bias as followed: we acknowledge that the selection of study participant could have been biased toward health-motivated individuals and this could have inflated estimates of HIV prevalence; nevertheless, we also think that because this group of health-motivated individuals are mixed with other healthy individuals in the community, there is not expected to be a significant HIV prevalence difference between these individuals and those in the general population.
The diagnostic algorithm used is a relative strength of the paper. It follows WHO guidelines for low-prevalence settings (<5%) and includes a well-described series of rapid and confirmatory assays, ensuring internal validity of HIV test results. However, the logistics of blood collection—5 mL samples in remote and hot environments are not sufficiently described. There is no information on sample transport, storage conditions, or cold chain management, which are crucial in determining test reliability in field conditions. Authors cou Blood samples (5 ml) were collected in the villages by trained phlebotomists and transported in insulated cool boxes with ice packs to maintain a temperature of 2–8 °C. They were first processed at the nearest referral hospital immediately after collection and subsequently transferred to the reference laboratory in Yaoundé for confirmatory analysis.ld have referenced other papers where the survey procedures have been completely documented, or put the details as supplementary material.
Answer: We added the following reference to strengthen this point:
“High HIV burden and recent transmission chains in rural forest areas in southern Cameroon, where ancestors of HIV-1 have been identified in ape populations”. Edoul G, Chia JE Guichet E, Montavon C, Delaporte E, Mpoudi Ngole E, Ayouba A, Peeters M.Infect Genet Evol. 2020 Oct;84:104358. doi: 10.1016/j.meegid.2020.104358. Epub 2020 May 18.PMID: 32439500
Perhaps the most important shortcoming of the methodology is the statistical analysis. The outcome of interest (HIV status) is a binary outcome (yes/no), yet the authors rely on Poisson and negative binomial regressions, which are designed for count data. While modified Poisson regression can be validly used for binary outcomes to estimate prevalence ratios (as per Barros and Hirakata, 2003), the authors do not explain their rationale or address critical assumptions. Their subsequent shift to negative binomial regression due to “overdispersion” is methodologically unjustified. Overdispersion in binary data is conceptually different from that in count data and typically points to issues like clustering or model misspecification. There is no evidence presented (e.g., Pearson chi-square or deviance/df ratios) to support this decision, nor is the dispersion parameter reported.
Using negative binomial regression for a binary outcome risks biased estimates and incorrect standard errors. It introduces assumptions (such as gamma-distributed heterogeneity) that are not applicable to binary outcomes. Standard practice in cross-sectional HIV seroprevalence studies is to use logistic regression or, where prevalence is low, modified Poisson regression with robust variance estimators. Log-binomial regression is another valid method but may face convergence issues. None of these alternatives are discussed.
Answer: Thanks for the comments. We have now modified the corresponding section of the methodology as follows: “Since this was a cross-sectional study and we aimed to estimate prevalence ratios (PRs) directly, the modified Poisson regression method may have been appropriate, because PRs are more interpretable than odd ratios (ORs) in cross-sectional studies (Barros and Hirakata[22]). However Ibáñez-Pinilla et al. [23] recently showed that a negative log-binomial model (i.e. negative binomial with log link) with robust variance outperforms the modified Poisson regression.”
In addition, Ibáñez-Pinilla et al. [23] shows that negative log-binomial models do not present with convergence issues, especially for high prevalence cases, and could be considered in cases of overdispersion and with greater precision and goodness of fit than the other models with robust variance.
- Ibáñez-Pinilla M, Villalba-Niño S, Olaya-Galán NN. Negative log-binomial model with optimal robust variance to estimate the prevalence ratio, in cross-sectional population studies. BMC Med Res Methodol. 2023 Oct 4;23(1):219. doi: 10.1186/s12874-023-01999-1. PMID: 37794385; PMCID: PMC105485899.
There is also no assessment of multicollinearity, interaction terms (e.g., age by sex or region), or model diagnostics such as AIC, BIC, or pseudo-R². This analytical approach undermines the validity of the reported adjusted prevalence ratios (e.g., aPR = 2.27 for the South region; aPR = 1.48 for females).
Answer: In the manuscript, we have already mentioned that, variables used for adjustment were those with p-values less than 0.05 at univariate analysis, these included age group, sex, region, and marital status. We have now included in Tables 1 and 2 model diagnostic metrics in terms of goodness of fit: Quasi Likelihood under Independence Model Criterion (QIC). We did not assess multicollinearity and interaction terms because there would have been too many combinations. In fact Age has 11 classes, Sex 2 classes, Region 4 classes, Marital status 4 classes. We also believe that trying to do so goes far beyond the scope of the journal and the paper; thus we kept the main effects for ease in communication.
In addition, the authors ignore clustering in the data. With 46 villages sampled, standard errors are likely deflated due to intra-cluster correlation. This could explain the perceived “overdispersion” and should have been addressed using generalized estimating equations (GEE) or multilevel modeling approaches (e.g., mixed-effects logistic regression). The lack of adjustment for clustering is a serious flaw.
Answer: Thanks for this important point. As suggested, we have now re-analysed the data using negative log-binomial model with generalized estimating equations (GEE)
Behavioral data reportedly collected in the questionnaire, such as sexual history, condom use, or number of partners, are not analyzed or even described in the results. This is a missed opportunity, particularly given the manuscript’s emphasis on “epidemiology” and “risk factors.” The authors should either include analyses of these variables or clarify their absence.
Answer: The administered questionnaire included behavioural data such as number of sexual partners, and HIV testing history. These variables were not analysed in detail in the present manuscript to maintain the focus on sociodemographic determinants. They will be explored further in subsequent analyses and publications.
Results
The results section is generally well organized and includes helpful tables and figures, but several inconsistencies and errors undermine their clarity. Table 1 presents demographic characteristics of participants, but the sex distribution (60.6% male) is unusual for HIV testing surveys, which typically oversample women. The median age of 31 years is plausible, but the interquartile range (22–42) suggests a youth skew, which appears at odds with the manuscript’s emphasis on older adult prevalence.
Answer: This is a good point. Our rural communities are still a bit conservative. As a consequence, women sometimes require permission to participate in such surveys from their male partner; this might limit their participation. In addition, their availability might also be limited by the labour intensiveness of their occupations (such as farming).
Figure 3 clearly shows the age-sex breakdown of HIV prevalence and successfully highlights peaks among women aged 35–44 and 55+, and men aged 45–54. However, the sample sizes for older age groups are small (often <10% per band), leading to potential instability in prevalence estimates. Confidence intervals should be added to illustrate this.
Answer: We acknowledge that the sample sizes for older age groups were small, and this may have limited the precision of prevalence estimates. We have added this point in the discussion section of the initial submission. In addition, we have included the 95% confidence intervals to tables and figures to reflect this uncertainty.
Table 2 presents adjusted prevalence ratios but does not clearly distinguish between crude and adjusted models. The p-values are inconsistently formatted (e.g., use of commas for decimals) and some categories are poorly labeled or duplicated (e.g., “50 or more” appears twice). There are also typographical errors throughout. Importantly, there is no model fit information reported (e.g., goodness-of-fit tests, AIC/BIC), nor are diagnostics provided. The authors also exclude 55 individuals with indeterminate results (~1%), but do not test whether their exclusion affects the findings.
Answer: Thanks. It is now fixed. The goodness of fit is now reported as Quasi Likelihood under Independence Model Criterion (QIC). Age has 11 classes, but we have split it in two categories (15-49, 50 or more) just to compare both groups, in order to show the usefulness of not excluding older adults. However, multivariate analysis used only age group with 11 categories to avoid redundancy. Before beginning the analysis, we assessed (not reported here) the effects of excluding 55 individuals with indeterminate results, no influence was detected.
Discussion
The discussion is the manuscript’s strongest section. It thoughtfully connects findings to the broader literature, such as the consistently elevated prevalence in the South region (5.2%), which aligns with previous studies suggesting a stable, long-standing epidemic in this area. The linkage of male vulnerability to mobility and trade routes is also well reasoned. The authors correctly critique the standard practice of excluding older adults (≥50) from surveillance and propose important hypotheses, such as survival bias among ART users or rural exodus returning older people to their home villages.
That said, the discussion overreaches in several places. It refers to a “mature and long-standing epidemic” without supporting incidence or temporal data. It attributes the education-HIV association (more educated individuals having higher prevalence) to vulnerability without considering that this may reflect reverse causality or residual confounding. Comparisons to urban DHS data are problematic because the current sample is rural and non-random. The discussion of HIV-1 subtype diversity is tangential and underexplored. Limitations are only partially acknowledged: selection bias is noted, but not quantified; the omission of behavioral variables and the use of flawed regression models are ignored. There is no mention of ART coverage or viral load status, both of which would help distinguish prevalent vs. undiagnosed cases. The discussion would benefit from a clearer acknowledgment of these limitations.
Answer: We welcome this suggestion. We added the limitations in the discussion section as follows:
Our results confirm marked regional disparities and higher prevalence among adults aged 35 years and above, particularly in women. However this study has several limitations including the non-random sampling design, volunteer-related selection bias, small sample sizes in certain subgroups, the lack of detailed analyses of sexual behaviors. In addition, clinical data (viral load, CD4 count) and ART coverage were not available to distinguish prevalent vs. undiagnosed cases.
Conclusions
The conclusions are concise and correctly call for surveillance systems to better capture older adults and underserved regions. However, they overstate the evidence by referring to “persistent drivers” without having measured any such mediating factors. The conclusions should align more closely with the limitations of the study, particularly the non-probability sampling and flawed analysis. Calls for inclusion in UNAIDS targets are valuable but should be framed with appropriate caveats about data quality.
Answer: Despite some limitations on sampling bias and size, our data underscore the need to expand epidemiological surveillance to include individuals aged 50 years and older and to conduct further research to better understand the determinants of transmission in these specific contexts.
Figures and Tables
Figure 2 (diagnostic algorithm) is well-designed but should include test sensitivity/specificity. Figure 3 effectively shows prevalence curves but lacks confidence intervals. Tables 1–2 need to be cleaned for typos, formatting consistency, and inclusion of 95% confidence intervals.
Answer: Done, thanks
Major revisions are necessary, particularly in the statistical approach, contextual framing, and interpretation of results. If addressed rigorously, the study could make a meaningful contribution to rural HIV epidemiology and policy.
Comments on the Quality of English Language
Thorough language editing should be performed
Answer: this was done.